# Two-component nematic superconductivity in 4Hb-TaS₂

I. Silber[1], S. Mathimalar[2], I. Mangel[3], A. K. Nayak[2], O. Green[1], N. Avraham [2],
H. Beidenkopf[2], I. Feldman[3], A. Kanigel[3], A. Klein [4,5], M. Goldstein [1],
A. Banerjee[6], E. Sela[1] & Y. Dagan [1] ✉

Most superconductors have an isotropic, single component order parameter and are well described by the standard (BCS) theory for superconductivity. Unconventional, multiple-component superconductors are exceptionally rare and are much less understood. Here, we combine scanning tunneling microscopy and angle-resolved macroscopic transport for studying the candidate chiral superconductor, 4Hb-TaS₂. We reveal quasi-periodic one-dimensional modulations in the tunneling conductance accompanied by two-fold symmetric superconducting critical field. The strong modulation of the in-plane critical field, $H_{c2}$, points to a nematic, unconventional order parameter. However, the imaged vortex core is isotropic at low temperatures. We suggest a model that reconciles this apparent discrepancy and takes into account previously observed spontaneous time-reversal symmetry breaking at low temperatures. The model describes a competition between a dominating chiral superconducting order parameter and a nematic one. The latter emerges close to the normal phase. Our results strongly support the existence of two-component superconductivity in 4Hb-TaS₂ and can provide valuable insights into other systems with coexistent charge order and superconductivity.

Superconductivity in metals and alloys is usually characterized by an isotropic order parameter. This isotropy stems from the short length scales of the charge screening and the electron-phonon interaction. On the other hand, unconventional superconductors have an anisotropic and possibly multiple-components order parameter, transforming non-trivially under the symmetry operation of the underlying crystal[1–3]. In practice, evidence for multi-component superconductivity has been reported in a few compounds, e.g., NbSe₂[4], Bi₂Se₃[5–9], and iron-based superconductors[10,11]. A unique situation emerges in chiral superconductors when the order parameter is a time-reversal-symmetry-breaking linear combination of degenerate representations[12]. Chiral superconductivity is even more elusive, with possible candidates being UPt₃[13–15], UTe₂[16–18] and Sr₂RuO₄[19]. The exact nature of the superconductivity in the latter is still under debate[20].

Recently, mounting evidence points out that 4Hb-TaS₂ hosts a chiral superconducting ground state[21–24]. However, clear proof for the two-component superconducting state is lacking, and the superconducting phase diagram is yet to be determined.

4Hb-TaS₂ is a naturally occurring van der Waals heterostructure comprised of alternating 1T- and 1H-TaS₂ layers (Fig. 1a), belonging to the P36/mmc hexagonal space group symmetry (#194). 1T-TaS₂ is considered a Mott insulator and a quantum spin liquid candidate[25–28]. 2H-TaS₂ is a superconductor with a critical temperature of 0.7 K[29]. 4Hb-TaS₂ undergoes a charge density wave (CDW) transition at $T = 315\,K$[30], which is clearly observed as an abrupt increase in the resistance (see Fig. 1b) accompanied by a five-fold reduction of the Hall number (see inset of Fig. 1b and Supplementary Fig. S1). Nevertheless, the system remains metallic and

[1]School of Physics and Astronomy, Tel - Aviv University, Tel Aviv 69978, Israel. [2]Department of Condensed Matter Physics, Weizmann Institute of Science, Rehovot, Israel. [3]Physics Department, Technion-Israel Institute of Technology, Haifa 32000, Israel. [4]Department of Physics, Faculty of Natural Sciences, Ariel University, Ariel 40700, Israel. [5]Department of Chemical Physics, The Weizmann Institute of Science, Rehovot 76100, Israel. [6]Department of Physics, Ben-Gurion University of the Negev, Beer-Sheva 84105, Israel. ✉e-mail: yodagan@tauex.tau.ac.il

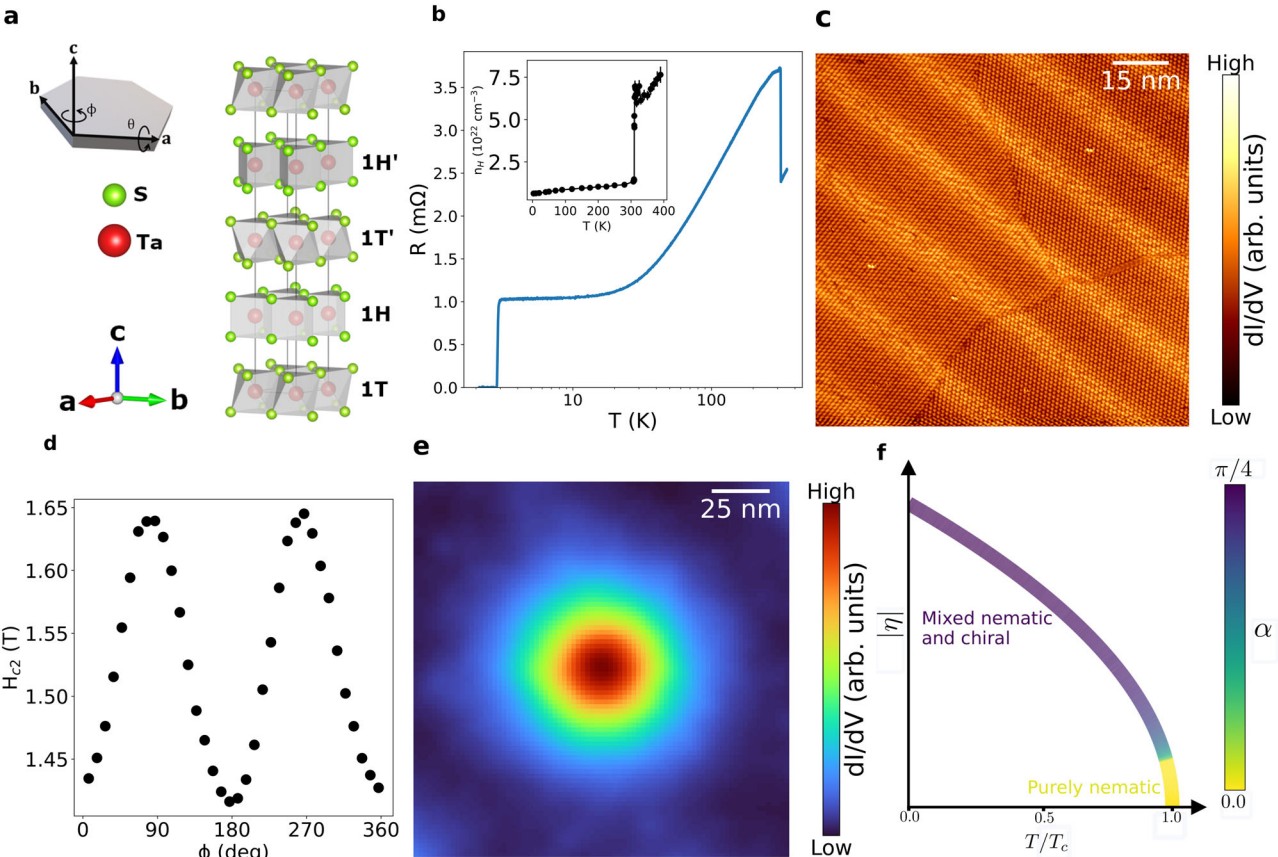

**Fig. 1 | Nematic and chiral superconductivity in 4Hb-TaS₂ and their origin.**
**a** Crystal structure of 4Hb-TaS₂ displaying alternately stacked quasi-2D layers of 1T-TaS₂ (1T) and half of the 2H-TaS₂ (1H) polymorphs. Inset: schematics of the hexagonal crystal, marking the crystal axes and the two directions of the field rotation in the experiment. **b** In-plane resistance as a function of temperature, while applying current in the ab-plane. At $T = 315$ K the charge density wave transition can be seen by the abrupt increase in the crystal resistance. The superconducting transition is observed at $T_c = 2.7$ K. Note the logarithmic temperature scale. Inset: Hall number as a function of temperature. The charge density wave phase transition at $T = 315$ K results in a strong reduction in Hall number. **c** STM topographic image,

where each dot represents a single site in the $\sqrt{13} \times \sqrt{13}$ CDW pattern. The stripes are clearly seen, and they extend across a CDW domain boundary. **d** The critical field (at $T = 2.5$ K) has a clear 2-fold symmetry as a function of the in-plane direction of the applied magnetic field. The current is applied at $\phi = 90°$. **e** Averaged STM image of a vortex core at $T = 0.38$ K, with 100 mT field applied out-of-plane. The density of states as a function of location, averaged over several vortices, is clearly isotropic. **f** The suggested phase diagram displays the variation of the order parameter $\eta$ with temperature. A nematic state is favored near $T_c$, whereas the order parameter becomes chiral at low temperatures. The nematic-chiral crossover is parameterized by the angle $\alpha$ (see the main text for details).

undergoes a superconducting transition at $T_c = 2.7$ K, as can be seen in Fig. 1b.

In this work we combine macroscopic magneto-transport and scanning tunneling microscopy (STM) to discover a nematic superconducting state in 4Hb-TaS₂ close to its superconducting transition temperature. We also propose a theoretical model to reconcile our findings with the observed breaking of time-reversal symmetry[21].

## Results

Each point in Fig. 1c is the 1T CDW super-cell, which dominates over the weaker 1H charge density modulations[22]. Surprisingly, we found modulations of the tunneling conductance in the form of highly oriented stripes with a separation of about $\approx 20$ nm. These stripes break the $C_6$ symmetry of the 4Hb-TaS₂ crystal.

Remarkably, this symmetry breaking is manifested in the superconducting state. When rotating the magnetic field in the basal plane, parallel to the TaS₂ planes of the sample, we observe significant twofold modulations of the critical field $H_{c2}$ (Fig. 1d). We link the minimal critical field with the direction of the stripes as both are parallel to the crystal's a-axis.

The stripes found by our STM data allow for nematic symmetry breaking in 4Hb-TaS₂, including its superconducting state. Therefore, we interpret the two-fold critical field modulation as the signature of a

multiple-component nematic superconducting order parameter, following Refs. 31–34.

Naively, an anisotropic $H_{c2}$ would result in an anisotropy of the vortex core[35]. However, the density of states in a vortex core at low temperatures is perfectly isotropic as seen in Fig. 1e. We argue that the observations of a nematic state and an isotropic vortex core can be reconciled by invoking a crossover from a chiral order parameter, which dominates at low temperatures, to a nematic one, appearing near $T_c$. The calculated zero field phase diagram is presented in Fig. 1f. Our model predicts a smaller anisotropy of $H_{c2}$ at low temperatures, as observed in the experiment and discussed further below.

We now focus on the characterization of the stripes and their possible origin. The stripes are well oriented across the entire sample and define a specific direction over a macroscopic length scale, roughly parallel to the crystal's a-axis, as seen in Fig. 2a. Their direction persists even when crossing a CDW domain wall, as demonstrated in Fig. 1c. We focus on the regions with abundant stripes and observe a quasi-periodic modulation of $\approx 19.6$ nm. In some rare regions no or only a single stripe is found, and the period is ill-defined.

We have performed local spectroscopic measurements on and off stripe, both in the normal and in the superconducting state well below $T_c$. For negative voltage biases, the density of states on a stripe differs from off a stripe (Fig 2b, c). This bias dependent difference suggests

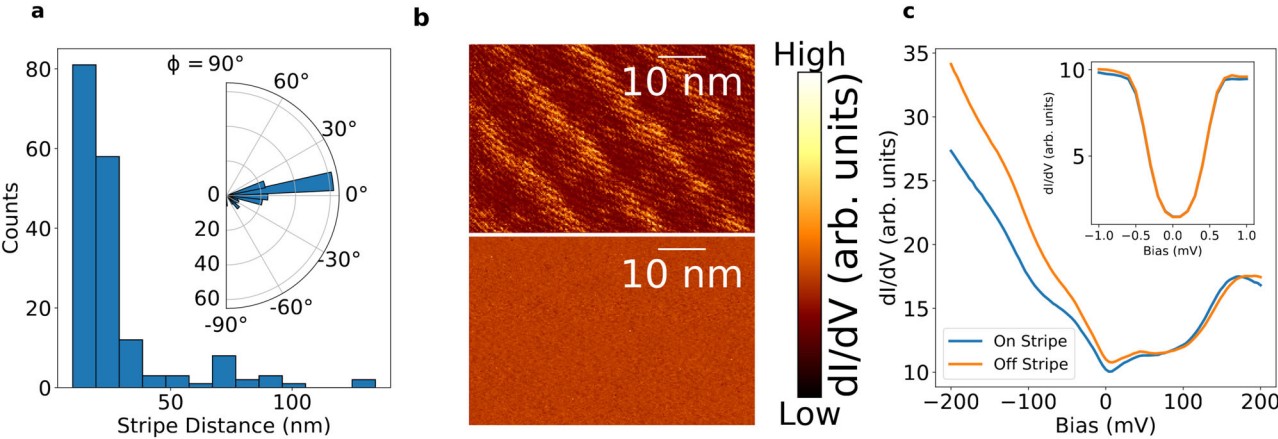

**Fig. 2 | Quasi-periodic conductance modulations in the form of stripes in 4Hb-TaS₂. a** Distance and angular distributions of the stripes show a stripe separation with a median $P_c = 19.6$ nm and a clear orientation, roughly parallel to the crystal's a-axis. **b** Differential conductance maps of the same field of view were measured at $E = -140$ meV (top), where the stripes are clearly seen, and at $E = 160$ meV (bottom), where the stripes are invisible. The absence of stripes at different energies shows that they are not simply a topographic modulation. **c** Spatially averaged dI/dV profiles measured on and off stripe, at 4 K. Inset: Low-bias differential conductance spectra measured at 0.4 K on and off stripe. The superconducting gap is unchanged, meaning the superconductivity is homogeneous throughout the sample.

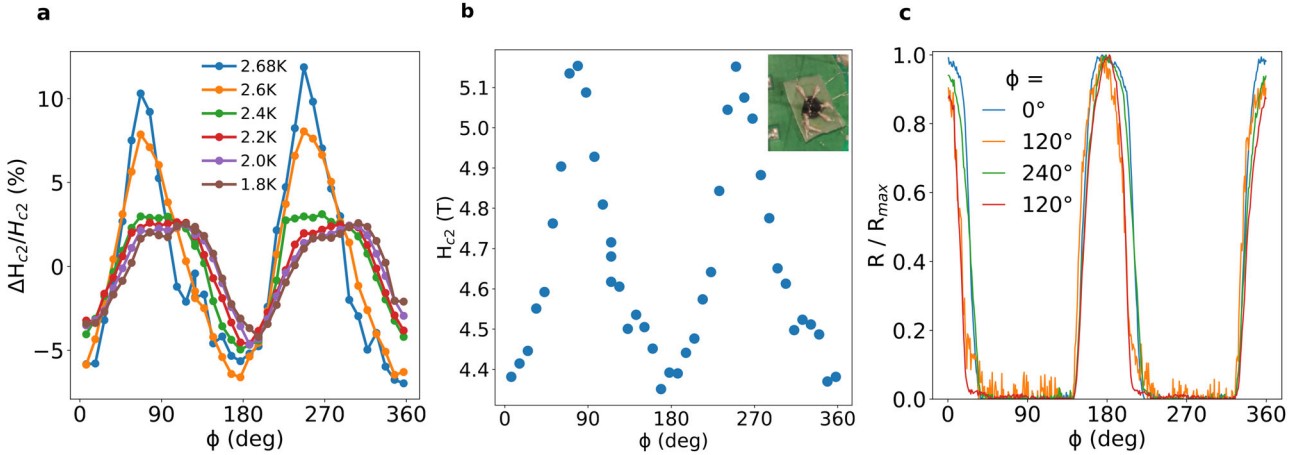

**Fig. 3 | Nematic superconductivity in 4Hb-TaS₂. a** Critical fields of the flake sample at various temperatures. The critical field has a pronounced two-fold symmetry, whose anisotropy is larger at temperatures closer to the critical temperature. The current is applied at $\phi = 210°$. **b** Similar measurement on a single crystal sample at $T = 2.1$ K, in which the crystal axes can be easily identified. The minimal critical field is observed when the field is applied parallel to the crystal's a-axis. The current is applied at $\phi = 20°$. Inset: micrograph of the single crystal sample with contacts attached. **c** Resistance measurements as a function of the applied magnetic field direction in the basal plane of the single crystal sample, for a constant magnetic field, $|H| = 3.1$ T, at $T = 2.1$ K. A peak in the resistance corresponds to a minimum in $H_{c2}$ at these angles. The various colors represent four different current directions, whose directions are specified in the legend. Clearly, the overall behavior is independent of the current direction. The two 120° data sets correspond to current leads put on two parallel faces of the crystal.

that the stripes represent a change in the local electronic density of states rather than just a topographic modulation. We further note that the superconducting gap itself, observed at low biases, remains the same on and off the stripes (Inset of Fig. 2c).

As mentioned above, when we rotate the magnetic field in the basal plane, a remarkable two-fold symmetry of the critical magnetic field $H_{c2}$ is detected, as seen in Fig. 3, and as detailed in Supplementary Fig. S6. The minimum of $H_{c2}$ is observed when the field is applied parallel to the crystal's a-axis, similar to the direction of the stripes. Surprisingly, the anisotropy of the critical field is smaller as the temperature is decreased from $T_c$ down to 1.8 K, and is much smaller at $T = 0.4$ K, hardly resolved from the data (see Supplementary Fig. S5).

## Discussion
The anisotropy of the critical field does not depend on the direction of the applied measurements current. On the large single crystals, we applied current at four different directions, and found the resistance as a function of the in-plane angle to have similar trend in all cases (see Fig. 3c). Moreover, the critical field variation in the thin flake did not depend on the direction of the current, as can be seen in Supplementary Fig. S2a. For more details about control experiments excluding spurious effects such as wobbling, see the Supplementary Information.

A straightforward explanation for the anisotropy of $H_{c2}$ is that it stems from a variation in the Josephson-vortex pinning due to the stripes. To test this conjecture, we model the stripes using a slow modulation of the inter-layer stacking configuration, yielding a variation of the perpendicular mass $m_z$ in a Ginzburg Landau (GL) theory along an in-plane coordinate, perpendicular to the observed stripes. Naturally, modulation of $m_z$ does not affect the round shape of the Abrikosov vortices in the plane. Solving the linearized GL equations, we generically find that the critical field is maximal parallel to the stripes. This is in contrast to our observations, ruling out this mechanism (see the Supplementary Information for details).

Moreover, slightly below the critical temperature we observe a clear in-plane rotational symmetry breaking, while at lower temperatures the superconducting state is much more isotropic (Supplementary Fig. S5), as seen in the round vortices (Fig. 1e) and the smaller variations in the in-plane critical field (Fig. 3a). We also note that 4Hb-TaS$_2$ is different than most multi-component superconductors, where the nematic behavior is observed throughout the entire temperature range of the superconducting domain[4–6].

Why does the imaged vortex core shows no anisotropy while the in-plane critical field strongly depends on the angle between the magnetic field and the crystal axis? We argue that this temperature dependence can arise between a competition between nematic and chiral superconductivity which we describe within a two-component GL theory[33]. This explanation describes our current results as well as the chiral superconducting ground state previously suggested in 4Hb-TaS$_2$[21,23,24].

The two-component order parameter can be written as $\eta = \eta \left( \begin{smallmatrix} \cos\alpha \\ \sin\alpha\, e^{i\gamma} \end{smallmatrix} \right)$, where $\eta$ is the amplitude of the superconducting order parameter, and the angles $\alpha$ and $\gamma$ parametrize the nematic and chiral phases. In a purely nematic state, either $\gamma = 0$ and $\alpha$ dictates the direction of nematicity and the resulting anisotropy, or $\alpha = 0$. On the other hand, the purely chiral order, $\gamma \to \pm \pi/2$ and $\alpha = \pi/4$, is isotropic. Deep in the superconducting state, the system is described by a chiral order parameter[21], leading to isotropic vortices as reported here.

The origin of the emergent stripes is not fully understood. A possible scenario is that stacking two dimensional layers with a small mismatch and different CDW patterns generates a finite uniaxial strain (see supplementary material for more information). Regardless of the exact microscopic origin, we include this experimentally observed symmetry breaking via a $\epsilon_{xx} - \epsilon_{yy}$ term in the GL theory. Solving the GL equations for different temperatures discussed in the supplementary material, we obtain that $\gamma = \pi/2$, although $\alpha$ varies with $T$. Fig. 1f shows that as the temperature increases and the order parameter is reduced, the strain term dominates to favor the nematic order near the critical temperature.

The transition we have modeled, which progresses from a fully nematic to a mixed chiral and nematic superconductivity, is anticipated to produce a slight kink in the specific heat, which would be very hard to detect experimentally. To simplify our model, we did not consider the possible presence of a dominant coexisting s-wave state or the contribution of excited quasi-particles, which would further diminish the observed kink in the specific heat (see the Supplementary Information for more details).

Following this model, our results can be understood as a competition between terms: at low temperatures, the order parameter is mostly chiral, in line with the isotropic vortex core in Fig. 1e and the almost isotropic $H_{c2}$ in Supplementary Fig. S5. At temperatures closer to T$_c$, the order parameter is mostly nematic, consistent with the observed two-fold angular dependence of the critical field seen in Fig. 3. This theory predicts that vortices gradually become anisotropic at higher temperatures as the order parameter becomes nematic.

To summarize, we find a microscopic stripe pattern in 4Hb-TaS$_2$ and link it to the macroscopic two-fold symmetry of the superconducting critical field. The large variation in the critical field, of about 20%, implies possible existence of a nematic order parameter close to the normal state, which is allowed due to the strain field. We exclude a simpler explanation of Josephson-vortex pinning along the stripes that predicts a minimum in $H_{c2}$ opposite to our findings. We use a theoretical model to reconcile our observations of nematic superconductivity with the previously-suggested chiral superconducting ground state in the system. Future study is needed to understand the details of the microscopic mechanism responsible for the competition between chiral and nematic states suggested by our phenomenology.

## Methods

### Sample growth
High-quality single crystals of 4Hb-TaS$_2$ were prepared using the chemical vapor transport method. Appropriate amounts of Ta and S were ground and mixed with a small amount of Se (1% of the S amount). The powder was sealed in a quartz ampule, and a small amount of iodine was added as a transport agent. The ampule was placed in a three-zone furnace such that the powder is in the hot zone. After 30 days, single crystals with a typical size of 5.0 mm × 5.0 mm × 0.1 mm grew in the cold zone of the furnace. The fact that the van der Waals stacking in 4Hb-TaS$_2$ arrives in the form of a single crystal allows for clean, homogeneous and large samples.

### Definition of the in-plane angle
We define zero in-plane angle as the direction parallel to the crystal's a-axis in both transport and STM. In large crystals, we identify the crystal facet to establish the angle. For STM data, the angle is determined using the imaged lattice. However, in thin flakes of the crystal, where crystal axes identification is challenging, we define zero in-plane angle as the angle of the minimal critical field, aligning with our measurements in large crystals.

### Transport measurements
Ohmic contacts were established by affixing platinum wires to the 4Hb-TaS$_2$ single crystals using silver epoxy or by depositing aluminum onto flake samples cleaved using Scotch tape. For the larger samples, a current of approximately 1mA was applied using a Keithley 6221 current source, and the voltage drop was recorded using a Keithley 2182A nanovoltmeter. Measurements on the flake samples were conducted employing a standard lock-in technique with a Stanford SR830 lock-in amplifier, applying a current of 250 nA at a frequency of 37 Hz. High-field scans (Supplementary Fig. S5) were performed at the National High Magnetic Field Laboratory (NHMFL). The resistance was measured using a Lakeshore 372 resistance bridge with a current of 0.316 mA. In all of the resistance measurements the current was applied along the basal (ab-plane) plane of the sample. The critical field was defined as the the field in which the resistance is half of the normal state resistance. To determine the Hall signal we averaged two current orientations and anti-symmetrized the results with respect to the applied magnetic field.

### Rotation platforms
Data in Figs. 1d, 3, and Supplementary Fig. S2 were collected in a cryogenic station with a one-axis rotator probe. The control experiment in Supplementary Fig. S3 used a piezo stack enabling two-axis rotation with continuous angle monitoring. High-field data in Supplementary Fig. S5 was obtained using a hybrid two-axis platform: an out-of-plane mechanical rotator and in-plane piezo rotation with a piezo actuator. The angle in the high-field facility was monitored by measuring three Hall sensors.

### STM measurements
The samples were kept in an inert nitrogen environment before transferring to the load-lock of the STM chamber, exposing the samples to ambient conditions for a minimal amount of time. The samples were cleaved in the STM load-lock at ultra-high vacuum at room temperature. The cleaved crystals were then inserted directly to the 4 K sample stage of the STM head for spectroscopic measurements. Commercial PtIr tips were treated on a freshly prepared Cu(111) single crystal for the stability of the tips and then used for the measurements. All the spectroscopic data were measured using standard lock-in techniques with a frequency of 733 Hz. In Fig. 1c the measurement was performed at a constant current mode with set value of 200 pA. We interpret the change at the signal as originating from a change in conductance, and not due to a change in topography, as the stripes' relative density of states changes with bias (Fig. 2b). In Figs. 1e, 2c the

voltage was set in the range of 0.1–10 mV depending on the bias scan range, and the AC modulation varied between 2 mV to 200 mV. To ensure reproducibility, three different cleaves were performed (with three different tips) giving similar spectra.

## Data availability

All data generated or analyzed during this study are included in this published article (and its supplementary information files).

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

## Acknowledgements

We thank M. Ben Shalom and J. Ruhman for useful discussions. A part of the work presented here was performed at the National High Magnetic Field Laboratory (NHMFL), supported by the National Science Foundation Cooperative Agreement no. DMR-1644779, the State of Florida and the DOE. We are indebted to David Graf for his valuable assistance at the high field laboratory. The experimental work performed at Tel Aviv university was supported by the Pazy Research Foundation Grant No. 326-1/22, Israel Science Foundation Grants No. 3079/20, 1711/23 and 476/22 the TAU Quantum Research Center and the Oren Family Chair for Experimental Physics. M.G. was supported by the Israel Science Foundation (ISF) and the Directorate for Defense Research and Development (DDR&D) Grant No. 3427/21 and by the US-Israel Binational Science Foundation (BSF) Grant No. 2020072. A.B. and E.S. acknowledge support from the European Research Council (ERC) grant agreement No. 951541, ARO (W911NF-20-1-0013). A. KL. acknowledges support from the Israel Science Foundation (ISF) and the Directorate for Defense Research and Development (DDR&D) Grant No 3467/21. The experimental work performed at the Technion was supported by the ISF Grant No. 1263/21. S.M present address is Department of Physics & Nanotechnology, SRM Institute of Science & Technology, Kattankulathur 603203, India.

## Author contributions

I.F. and A.KA. prepared the 4Hb-TaS$_2$ single crystals. S.M., A.K.N, N.A., and H.B. performed and analyzed the STM measurements. M.G., A.B., E.S., and A.KL. proposed the theoretical explanation and performed the calculation of the phase diagrams, I.S., O.G., and I.M. performed the transport measurements. Y.D. conceived the experiment. All authors discussed the findings, wrote the manuscript and reviewed its final version.

## Competing interests

The authors declare no competing interests.
