## [Peer Review File · Nature Communications]

Two-Component Nematic Superconductivity in 4Hb-TaS₂REVIEWER COMMENTS

Reviewer #1 (Remarks to the Author):

The authors study the 4Hb-TaS₂, combining low-temperature transport measurement and scanning tunneling microscopy (STM). The topic is interesting and timely, and the data-presented is of high quality. Before recommending its publication, I have a few comments for the authors to consider, that could potentially enhance the clarity and impact of this work. Specifically:

1. Figure 1d, and figure 3. Would it be possible to include multiple curves taken at different temperature near T_c? It would make the study more comprehensive to see the qualitative trend reproduced at various temperature, and how the quantitative magnitude depends on the temperature.
2. Figure 1e. Would it be possible for the author to include a SI section, showing the STM image before averaging? Averaging does tend to make images more isotropic, and it would be good for the reader to see the consequence of averaging by comparing the "before and after", and that it primarily suppresses measurement noise on top of what was intrinsically isotropic.
3. The samples seem to be raw crystal fresh out of the furnace, with ~ 0.1 mm thickness. I have two comments regarding the material that might make the work more relevant to the 2D material community.

(1) Could the author consider exfoliating the material, performing the same experiment and studying its thickness dependence? This in my opinion will significantly enhance the impact of the paper, as the electronic bandstructure of vdW material is known to drastically change (and often becoming more exotic) as it approaches 2D or quasi-2D limit. If multiple thickness (particularly atomically-thin) is beyond the scope of the work, even one more set of data with sample of tens of nm thickness would be a good addition (and is typically easy to obtain for most vdW materials).

(2) These materials are known to be more chemically-reactive, compared to graphene and semiconducting TMDs. Over time, realistic surface oxidation/degradation can occur. While the bulk transport properties should be largely insensitive to it, the STM data can be much more vulnerable to it. Could the author elaborate for the STM data, what is the time elapsed after sample is grown and before it is measured, and how was the sample stored/transferred in between? Could the author include in SI, STM data performed on the same sample surface, taken at different time? Is there a noticeable change of STM results, and what is the qualitative time scale? This information would not only be relevant in interpreting the results already presented in the paper (is the result a measurement of pristine or compromised surface), but of interest and guidance for follow-up works on this material.

Reviewer #2 (Remarks to the Author):

The authors measured the STM image, tunneling spectrum, vortex pattern, and in-plane resistivity versus rotation of in-plane magnetic field for a multiband CDW superconductor 4Hb-TaS₂ (T_c=2.7K). They found a nematic pattern in the STM image, the in-plane resistivity shows a twofold oscillation (at T=2.5K) versus the angle of the in-plane magnetic field, but the vortex image measured at 0.38K show an isotropic round shape. They then explain with a reconciled picture of a crossover from chiral to nematic superconductivity. I find the observations are quite trivial, which can be easily understood without the need of incorporating a sophisticated picture. First the angle-resolved macroscopic transport for a twofold symmetric feature in the flux flow state is not strange, especially the H_{c2} determined in this way only changes from about 4.4 to 5.2. With an in-plane current and in-plane magnetic field, in the flux flow region, some time it is quite easy to see a certain amount of pseudo-effect of twofold symmetry. Even there is a twofold symmetry, one cannot rule out the influence of the CDW order on the electronic properties, such as the Fermi velocity and Fermi surface. Given the slight anisotropy of critical field determined at 2.5K (very close to the T_c=2.7K), the vortex image measured at about 0.38K may still look quite isotropic, because the coherence length is related to the critical field in a square-root relation (based on GL theory). One can estimate how small the anisotropy would be if taking the slight anisotropy of the critical field into account. Thus I would hope the proposed fancy

picture is true, but the data do not support the conclusion at all. I cannot make a positive recommendation for the work.

Reviewer #3 (Remarks to the Author):

I have reviewed the manuscript NCOMMS-22-37257-T "Chiral to Nematic Crossover in the Superconducting State of 4Hb-TaS₂" by I. Silber et al. submitted to Nature Communications.

In this manuscript, the authors report combined investigation of superconductive properties in 4Hb-TaS₂ based on transport, scanning tunneling microscopy and spectroscopy (STM/S) and phenomenological theories. From transport, the authors found that this material exhibit two-fold anisotropy of in-plane upper critical field (H_{c2}). The nematic anisotropy divided by the mean H_{c2} seems to be stronger at higher temperature. In contrast, STM/S measurements reveal isotropic vortex-core structure. These two apparently contradicting results are explained by assuming a two-component superconducting (SC) order parameter: the order parameter couples so that it forms a nematic state at higher-temperature regime very close to $T_c = 2.7$ K whereas it forms a chiral state in a lower-temperature region.

The target material 4Hb-TaS₂ is one of the recently-discovered interesting superconductors and has been attracting much attention. The present manuscript proposes an interesting possibility on this material, but the main conclusion is based on the two apparently contradicting results. Moreover, I have technical concerns on the anisotropy in H_{c2} . Altogether, I do not recommend publication of the manuscript in Nature Commun.

Below I list my detailed comments on each issue.

(1) As I already mentioned above, one issue on this manuscript is that the main conclusion is based on two experiments that are apparently contradict with each other. This issue can be resolved by providing experimental results connecting between the two. For example, if nematic state is favored only at high temperatures, why the authors do not measure vortex-core structure at elevated temperatures to try to see nematic deformation of vortex cores? In my opinion, such firm experimental evidences are necessary for publication in high-standard journals.

(2) Moreover, for the in-plane H_{c2} anisotropy, possible field misalignment (i.e. tilted field-rotation plane with respect to the actual crystalline ab plane) cannot be completely excluded. This is because that 4Hb-TaS₂ is highly quasi-two-dimensional material with out-of-plane H_{c2} anisotropy of around 20 according to Ref.19. Then, assuming a GL anisotropic effective mass model, 1deg field misalignment can results in 5.5% extrinsic anisotropy in measured H_{c2} . If the H_{c2} behaves like the Tinkham's thin-film model, the misalignment effect can be even larger. Although the authors performed control experiments shown in Fig.S2, I think that this is not enough for the present case because it is difficult to assume perfect alignment among sample, sample state, rotator, and magnetic field (I have additional concerns as mentioned below). Notice that, for Bi₂Se₃-based nematic superconductors, out-of-plane anisotropy is as small as 2-3. Then experiments using a one-axis rotator can provide meaningful results. But for the present material having anisotropy of 20, special care must be taken. In my opinion, the only way to directly proof the absence of field-misalignment is, by using two-axis rotator, to check field misalignment from the out-of-plane rotation.

(3) If there's a transition from nematic to chiral SC states, there must be some indication, e.g. in specific heat and in the shape of $H_{c2}(T)$ curve. Are there any such supporting evidence for the scenario?

(4) In the normal-state magnetoresistance, are there any two-fold anisotropy? If yes, then this could

affect the H_{c2} data if H_{c2} is defined as the field where R/R_{normal} reaches certain values.

(5) For the control experiment in Fig.S2, I am a bit confused that the shift in the H_{c2} minimum or maximum positions are around 60 deg, although the authors claim that there's a shift of around 90 deg. This questions the validity of the claim. For this data, more accurate claim should be made. The shift of the min/max of $H_{c2}(\phi)$ should be evaluated more accurately by using certain fittings, and the rotation angle should be evaluated accurately from a photo of the setting.

(6) For the H_{c2} measurements at 0.4 K (Fig. S4), the authors' argument "We have found a much weaker anisotropy of H_{c2} , hardly resolved from the data, as seen in Fig. S4." may not be relevant because the data only covers the angle range of 0-100 deg. To claim the strength of the two-fold anisotropy, one needs data covering a range of at least 180-deg. Also, in relation to my comment (2), there's a possibility that the two-fold anisotropy is much weaker in this data merely because the field alignment in this measurement is much better than the other data since two-axis rotator is used in this measurement at NHMFL.

(7) Overall, many experimental informations are missing.

(a) How is the in-plane angle ϕ defined precisely? What is its relation to the crystalline a axis? For all data throughout transport and STM/S, is ϕ defined in a same way?

(b) For the data in Fig.1b, what is the current direction?

(c) For Fig.1e, what is the field direction (presumably c -axis??) and magnitude? Where is the a axis direction?

(d) For Figs. 1d, 3a, 3b, 3c, S2a, S2b, S4, S5, direction of the current in terms of ϕ should be mentioned.

(e) The authors mention "an in-plane crystal axis." What exactly does this mean (presumably the a axis)? Please add clear definition.

(8) Add explanation on the lattice symmetry (space group) of this compound, both above and below the CDW transition.

(9) In the introduction, Sr_2RuO_4 should be added as a chiral-superconductor candidate.

(10) " H_c " in Figs. 1d, 3a, 3b, S2a, S2b, S4 should be " H_{c2} "

(11) " ΔH_c " in Figs. 3a and S2a should be " $\Delta H_c/H_{c2}$ " or something like that.

(12) For the ϕ -dependence data, it is better to put the ticks in the horizontal axis every 90 deg, not 100 deg, so that one can analyse the asymmetry easily.

Response letter manuscript NCOMMS-22-37257-T

To the Editor, Nature Communications

We are hereby resubmitting our manuscript NCOMMS-22-37257-T "*Chiral to Nematic Crossover in the Superconducting State of 4Hb-TaS₂*," to Nature Communications. First, we would like to thank the Reviewers for reading the paper thoroughly and for the useful comments and suggestions that helped us improve our manuscript.

We are happy that the Reviewers found the topic timely and important and the data interesting. Reviewer #1 wrote that "*The topic is interesting and timely, and the data presented is of high quality*" Reviewer #3 wrote that "*The target material 4Hb-TaS₂ is one of the recently-discovered interesting superconductors and has been attracting much attention. The present manuscript proposes an interesting possibility on this material.*"

In the new version of the paper, we include more data on the temperature dependence of the anisotropy, theoretical modeling of the heat capacity, and more explanations and clarifications to address the remarks and criticisms of the Reviewers. The changes are marked in red color at the revised version.

Below we address the Reviewers' comments and remarks point by point. With this we believe our manuscript is now appropriate for publication in Nature Communications.

Reviewer #1:

The authors study the 4Hb-TaS₂, combining low-temperature transport measurement and scanning tunneling microscopy (STM). The topic is interesting and timely, and the data presented is of high quality.

We thank the Reviewer for finding our results interesting and timely.

Before recommending its publication, I have a few comments for the authors to consider, that could potentially enhance the clarity and impact of this work. Specifically:

1. Figure 1d, and figure 3. Would it be possible to include multiple curves taken at different temperature near T_c? It would make the study more comprehensive to see the qualitative trend reproduced at various temperatures, and how the quantitative magnitude depends on the temperature.

We thank Reviewer #1 for their suggestion. We added a new temperature set. The new data clearly show that the anisotropy changes rapidly as the temperature is lowered below T_c. Since both the in-plane H_{c2} and the out of plane one are linear in temperature, this strong temperature dependence also rules out external sources to the effect, such as probe wobble or sample misalignment, as discussed below.

2. Figure 1e. Would it be possible for the author to include a SI section, showing the STM image before averaging? Averaging does tend to make images more isotropic, and it would be good for the reader to see the consequence of averaging by comparing the "before and after", and that it primarily suppresses measurement noise on top of what was intrinsically isotropic.

We thank the Reviewer for the recommendation. We have added a supplementary section ("Averaging procedure of the vortex core") that displays the raw data, smoothed data, and

the regions that were used to calculate the averaged vortex core. The raw and smoothed data are quite isotropic already.

3. The samples seem to be raw crystal fresh out of the furnace, with ~ 0.1 mm thickness. I have two comments regarding the material that might make the work more relevant to the 2D material community.

(1) Could the author consider exfoliating the material, performing the same experiment and studying its thickness dependence? This in my opinion will significantly enhance the impact of the paper, as the electronic band structure of vdW material is known to drastically change (and often becoming more exotic) as it approaches 2D or quasi-2D limit. If multiple thickness (particularly atomically-thin) is beyond the scope of the work, even one more set of data with sample of tens of nm thickness would be a good addition (and is typically easy to obtain for most vdW materials).

We agree with the Reviewer that a thin flake is an important addition to the data measured on a large thick crystal. We have included measurements on the aforementioned crystal and on a thin flake (~100nm, estimated by optical microscopy). The data are similar for the thin flake and for the bulk crystal.

(2) These materials are known to be more chemically-reactive, compared to graphene and semiconducting TMDs. Over time, realistic surface oxidation/degradation can occur. While the bulk transport properties should be largely insensitive to it, the STM data can be much more vulnerable to it. Could the author elaborate for the STM data, what is the time elapsed after sample is grown and before it is measured, and how was the sample stored/transferred in between? Could the author include in SI, STM data performed on the same sample surface, taken at different time? Is there a noticeable change of STM results, and what is the qualitative time scale? This information would not only be relevant in interpreting the results already presented in the paper (is the result a measurement of pristine or compromised surface), but of interest and guidance for follow-up works on this material.

The samples were kept in an inert nitrogen environment before transferring them to the load-lock of the STM chamber. The sample was exposed to an ambient atmosphere for a few minutes during this transfer. The sample was cleaved in UHV in the STM preparation chamber (at room temperature) to expose a pristine surface for tunneling imaging and spectroscopy. Three different cleaves were performed (with three different tips) to ensure reproducibility, giving similar spectra (see Figure 1 in the response letter). We have better clarified this issue in the methods section.

Figure 1 - two tunneling spectra measured over two different cleaves, using different tips. The superconducting gap is reproducible.

Reviewer #2:

The authors measured the STM image, tunneling spectrum, vortex pattern, and in-plane resistivity versus rotation of in-plane magnetic field for a multiband CDW superconductor 4Hb-TaS₂ ($T_c=2.7K$). They found a nematic pattern in the STM image, the in-plane resistivity shows a twofold oscillation (at $T=2.5K$) versus the angle of the in-plane magnetic field, but the vortex image measured at 0.38K show an isotropic round shape. They then explain with a reconciled picture of a crossover from chiral to nematic superconductivity.

We thank the Reviewer for the thorough reading of the paper.

I find the observations are quite trivial, which can be easily understood without the need of incorporating a sophisticated picture. First the angle-resolved macroscopic transport for a twofold symmetric feature in the flux flow state is not strange, especially the H_{c2} determined in this way only changes from about 4.4 to 5.2. With an in-plane current and in-plane magnetic field, in the flux flow region, some time it is quite easy to see a certain amount of pseudo-effect of twofold symmetry.

If we understand correctly, the Reviewer proposes a flux-flow picture in which the vortices flow due to the in-plane currents. In this case, the flux flow resistance should strongly depend on the current direction with respect to the magnetic field. However, in Figures 3c and S2a, we clearly demonstrate that the angular dependence of the critical field is independent of the in-plane current direction.

In addition, a more complex picture of in-plane Josephson vortices could have explained an anisotropy. In this picture, Josephson vortices resulting from the in-plane magnetic field eventually break superconductivity when the field is strong enough. The anisotropy can arise if these vortices are pinned parallel to the stripes. However, this is the scenario we discuss and negate on page 3. In fact, for such a scenario, we expect the maximum critical field to be parallel to the stripes and not perpendicular to them, as we observe.

In summary, we believe that the anisotropy probed by the transport stems from the anisotropic intrinsic superconducting properties. This anisotropy is significant only near T_c and can be probed by a parallel critical field.

Even if there is a twofold symmetry, one cannot rule out the influence of the CDW order on the electronic properties, such as the Fermi velocity and Fermi surface.

We fully agree that the charge density wave order can influence the electronic properties. In the supplementary section "Proposed Origin of the Stripes," we interpret the stripes imaged by STM as a result of the unique CDW in this system that comprises 1H and 1T layers having different CDW orders as bulk materials. In addition, we clearly see in figure 1 that the CDW significantly affects the Hall coefficient, suggesting partial reconstruction of the Fermi surface. On the other hand, the strong anisotropy is observed only below T_c at temperatures much lower than the charge density wave transition. See also our response to point (3) by Reviewer #3.

Given the slight anisotropy of critical field determined at 2.5K (very close to the $T_c=2.7K$), the vortex image measured at about 0.38K may still look quite isotropic, because the coherence length is related to the critical field in a square-root relation (based on GL theory). One can estimate how small the anisotropy would be if taking the slight anisotropy of the critical field into account. Thus I would hope the proposed fancy picture is true, but the data do not support the conclusion at all. I cannot make a positive recommendation for the work.

Taking different line cuts at various angles around the averaged vortex core, we see no change. Using the standard scaling, a $\sim 20\%$ variation of the critical field should result in about $\sim 10\%$ variation of the in-plane coherence length. We estimate that we could detect such an anisotropy in the vortex core. The measured variation is clearly smaller than 10% , as can be seen by plotting a gaussian with 1.1 the full-width half maximum:

Figure 2 - the measured data in the manuscript, overlaid with a simulation of 10% increase in the vortex core.

Reviewer #3:

I have reviewed the manuscript NCOMMS-22-37257-T "Chiral to Nematic Crossover in the Superconducting State of 4Hb-TaS2" by I. Silber et al. submitted to Nature Communications.

In this manuscript, the authors report combined investigation of superconductive properties in 4Hb-TaS2 based on transport, scanning tunneling microscopy and spectroscopy (STM/S) and phenomenological theories. From transport, the authors found that this material exhibit two-fold anisotropy of in-plane upper critical field (H_{c2}). The nematic anisotropy divided by the mean H_{c2} seems to be stronger at higher temperature. In contrast, STM/S measurements reveal isotropic vortex-core structure. These two apparently contradicting results are explained by assuming a two-component superconducting (SC) order parameter: the order parameter couples so that it forms a nematic state at higher-temperature regime very close to $T_c = 2.7$ K whereas it forms a chiral state in a lower-temperature region.

The target material 4Hb-TaS2 is one of the recently-discovered interesting superconductors and has been attracting much attention. The present manuscript proposes an interesting possibility on this material

We thank the Reviewer for reading our manuscript. We agree that the superconducting properties of 4Hb-TaS2 are exciting.

but the main conclusion is based on the two apparently contradicting results. Moreover, I have technical concerns on the anisotropy in H_{c2} . Altogether, I do not recommend publication of the manuscript in Nature Commun.

In this paper, we show both STM and transport data. On the one hand, the shape of the vortex core and the strong anisotropy found in the transport close to T_c are apparently contradicting. On the other hand, the STM data found the microscopic origin for the breaking of the 6-fold symmetry of the crystal into the 2-fold symmetric structure, namely, the stripes. In that sense, the probes are complementary and point to the same conclusion. We believe the robust anisotropy observed in the transport and the connection to the microscopic STM observations are intriguing and deserve publication in Nature Communications.

On the theory side, the symmetry breaking observed by the STM is interpreted by us as a result of strain, allowing the anisotropic superconducting order parameter. We hope that our response below will convince the Reviewer that we can reconcile the round vortices observed at zero field and very low temperatures and the anisotropic critical field observed close to superconducting critical temperature.

Below I list my detailed comments on each issue.

(1) As I already mentioned above, one issue on this manuscript is that the main conclusion is based on two experiments that are apparently contradict with each other. This issue can be resolved by providing experimental results connecting between the two. For example, if nematic state is favored only at high temperatures, why the authors do not measure vortex-core structure at elevated temperatures to try to see nematic deformation of vortex cores? In my opinion, such firm experimental evidences are necessary for publication in high-standard journals.

We fully agree that it would be beneficial to image the vortex core at temperatures close to T_c . However, this is a very challenging experiment since the pinning force is tiny at that temperature range, and the Abrikosov lattice is unstable at low vortex density. Due to this

instability, it is almost impossible to image the vortex core noninvasively. We also note that the transport probed the in-plane anisotropy while applying a strong in-plane field. Imaging out-of-plane vortices while applying an in-plane field is even more complicated.

While we cannot measure the vortex core close to T_c , we added more transport data showing the evolution of the anisotropy with temperature. We clearly see that the strongest anisotropy is observed very close to T_c . The anisotropy quickly saturates as the temperature is lowered. Our simplified model can describe strong temperature dependence near T_c .

(2) Moreover, for the in-plane H_{c2} anisotropy, possible field misalignment (i.e. tilted field-rotation plane with respect to the actual crystalline ab plane) cannot be completely excluded. This is because that $4Hb-TaS_2$ is highly quasi-two-dimensional material with out-of-plane H_{c2} anisotropy of around 20 according to Ref.19. Then, assuming a GL anisotropic effective mass model, 1deg field misalignment can result in 5.5% extrinsic anisotropy in measured H_{c2} . If the H_{c2} behaves like the Tinkham's thin-film model, the misalignment effect can be even larger. Although the authors performed control experiments shown in Fig.S2, I think that this is not enough for the present case because it is difficult to assume perfect alignment among sample, sample state, rotator, and magnetic field (I have additional concerns as mentioned below). Notice that, for Bi_2Se_3 -based nematic superconductors, out-of-plane anisotropy is as small as 2-3. Then experiments using a one-axis rotator can provide meaningful results. But for the present material having anisotropy of 20, special care must be taken. In my opinion, the only way to directly proof the absence of field-misalignment is, by using two-axis rotator, to check field misalignment from the out-of-plane rotation.

A constant tilt resulting in a small out-of-plane field cannot explain the effect. It will merely result in a constant reduction of the critical field due to the constant out-of-plane component. Rotation in another plane (tilted with respect to the plane of the sample) can result in an out-of-plane wobble and an artificial two-fold change in H_{c2} . However, this must be detected in the two-flake control experiment (fig S2). Since both flakes are mounted together on the same platform, a possible wobble should be detected in both, irrespective of their relative in-plane orientation. This contrasts with what we observe: the important axis is the crystal axis and not the absolute angle of the probe. We stress that in this experiment, we do not assume any alignment. We only look for a wobble of the probe platform itself.

In addition, we used two different rotator systems (fig 3a and fig 3b), showing similar results, further supporting that this is not a particular probe wobbling.

Finally, we note that the $H_c(\phi)$ curves are non-trivially temperature dependent. The curves are strongly anisotropic near T_c . The anisotropy quickly decreases as the temperature is decreased and saturates around 2K. It is not easy to generate such anisotropy with geometrical factors. The geometry of the experiment should not be temperature dependent and definitely should not saturate at low temperatures, as we present in the new data included in the revised manuscript.

(3) If there's a transition from nematic to chiral SC states, there must be some indication, e.g. in specific heat and in the shape of $H_{c2}(T)$ curve. Are there any such supporting evidence for the scenario?

We thank the Reviewer for raising this point.

We made many more calculations following the comments of Reviewer #3 to reconcile the temperature dependence of the heat capacity with the anisotropic H_{c2} . We have added the

new calculation and discussion in the supplementary under "Specific heat from the Ginzburg Landau theory".

Within this analysis we indeed find that if the transition is from a purely-nematic to a mixed nematic-chiral, the corresponding non-analytic dependence of the order parameter on the temperature would be accompanied by a kink in the specific heat. However, for relevant values of the strain parameter λ_η that kink could be very weak. Moreover, if we add additional terms, allowed by symmetry, to the Ginzburg-Landau free energy, such as the term proportional to B_z which favors chirality, the transition at T^* becomes a crossover, smoothing out the kink. These points are exemplified here in Figure 3:

Figure 3 - variation of the specific heat due to chiral to nematic crossover. (left) - the calculated specific heat for various strain parameter, without the added B_z parameter. (right) - with a finite but small B_z parameter the minute kink at T^* becomes even smaller.

For simplicity, we did not consider the existence of a constate S-wave component nor the excited quasi-particles. Those will dominate the heat capacity at T_c and will further reduce the kink at T^* . In any case, the predicted small features at T^* might be too small to detect in our measurements.

(4) In the normal-state magnetoresistance, are there any two-fold anisotropy? If yes, then this could affect the H_{c2} data if H_{c2} is defined as the field where R/R_{normal} reaches certain values.

We have carefully measured the anisotropy of the normal state magnetoresistance at various temperatures. We observe a tiny effect of about 0.4% very close to T_c (see Figure 4). This tiny effect can be due to the misalignment of the probe or due to small normal state anisotropy that we are still studying. However, this tiny effect cannot give rise to a large change in the critical field.

Figure 4 – the measured small anisotropic magneto resistance of the single crystal sample, with field of 14T rotated in the plane of the sample.

We also note that we get similar trends if we estimate the critical field as the field at which the resistance derivative w.r.t field is maximal rather than half of the normal state resistance. We chose to use the half resistance criterion as it is much more robust to measurement noise and numerical differentiation artifacts.

Figure 5 - critical field as a function of the in-plane angle. Blue - critical field deduced by half of the normal state resistance. Orange - critical field deduced by maximal derivative of the resistance with respect to the field.

(5) For the control experiment in Fig.S2, I am a bit confused that the shift in the H_{c2} minimum or maximum positions are around 60 deg, although the authors claim that there's a shift of around 90 deg. This questions the validity of the claim. For this data, more accurate claim should be made. The shift of the min/max of $H_{c2}(\phi)$ should be evaluated more accurately by using certain fittings, and the rotation angle should be evaluated accurately from a photo of the setting.

We agree with the Reviewer that this point should be clarified. For all the samples in which we can reliably deduce the crystal "a" axis (i.e., thick single-crystals), the minimum in the critical field is parallel to that axis. Since there is a two-fold symmetry, only the minimum critical field can correlate with a certain crystallographic direction. The maximal critical field can shift by about 30 degrees from sample to sample (in relation to the direction of the "a" axis). Therefore, we use the minimum of the critical field as our reference point.

We calculate the position in which the minimal critical field occurs by fitting a local second-degree polynomial and finding its minimum analytically. The relevant locations, displayed in red, are: $[315^\circ, 135^\circ]$ for one cleave, and 204° for the other. If we estimate the relative rotation of the two cleaves, we get $[111^\circ, 69^\circ]$, and by averaging the two, we calculate a relative rotation of about 90 degrees, roughly the angle between the two visible axes in these crystals.

Figure 6 - calculation of the relative rotation between the two cleaves of the same single crystal. Blue - one of the cleaves. Orange - the other cleave, rotated by about 90 degrees with respect to the first. Red- Fits to second degree polynomials and their interpolated local minima.

(6) For the H_{c2} measurements at 0.4 K (Fig. S4), the authors' argument "We have found a much weaker anisotropy of H_{c2} , hardly resolved from the data, as seen in Fig. S4." may not be relevant because the data only covers the angle range of 0-100 deg. To claim the strength of the two-fold anisotropy, one needs data covering a range of at least 180-deg. Also, in relation to my comment (2), there's a possibility that the two-fold anisotropy is much weaker in this data merely because the field alignment in this measurement is much better than the other data since two-axis rotator is used in this measurement at NHMFL.

The two-axes rotator is available only in NHMFL. However, we could not perform a full rotation due to the limited magnet time. Therefore, the error bar is large; hence, we can merely set an upper limit on the variation amplitude of H_{c2} .

Nevertheless, we are positive that the effect measured on a single-axis rotator is not an artefact. First, we show the two-crystal rotation (Figure S2), and second, the temperature dependence cannot be due to probe or sample misalignment, as explained above. Finally, we note that the average value of the in-plane critical field fits the carefully measured in-plane critical field, see Ribak et al. (Ref. 21 in the revised manuscript). Therefore, any misalignment must be small, since an out-of-plane field component strongly affects the critical field.

(7) Overall, many experimental informations are missing.

We thank the Reviewer for this comment. As listed below, we have added explanations that will hopefully aid the future readers to fully understand our results.

(a) How is the in-plane angle ϕ defined precisely? What is its relation to the crystalline a axis? For all data throughout transport and STM/S, is ϕ defined in a same way?

We define zero in-plane angle $\phi = 0$ as the angle in which the field is parallel to the a-axis of the crystal, both in transport and in STM. In large samples (Fig 3d, 3c, S4, S5), we identified the crystal facet to mark the angle. In the STM, the angle was defined in relation to the imaged lattice. However, in thin flakes of the crystal (Fig. 1d, Fig 3a, Fig S2a), we could not identify the crystal axis, and we chose $\phi = 0$ as the angle of the minimal critical field. We have added a relevant Method section to clarify this.

(b) For the data in Fig.1b, what is the current direction?

The current is applied in the crystal's basal (ab) plane, approximately parallel to the "a" axis of the crystal in Fig.1b. In all the transport measurements, the current was applied in the basal plane of the crystals. We have added this clarification to the Methods section.

(c) For Fig.1e, what is the field direction (presumably c-axis??) and magnitude? Where is the a axis direction?

In Fig. 1e, a field of 100mT was applied out-of-plane (along c-axis). We added that in the manuscript and the supplementary section regarding the averaging process. The direction of the "a" axis is irrelevant as the vortex core is fully isotropic.

(d) For Figs. 1d, 3a, 3b, 3c, S2a, S2b, S4, S5, direction of the current in terms of ϕ should be mentioned.

We added a paragraph discussing the independence of the effect on the current direction, and we added the direction of the applied current in terms of ϕ for these figures.

(e) The authors mention "an in-plane crystal axis." What exactly does this mean (presumably the a axis)? Please add clear definition.

At room temperature, the a and b axes of the crystal are equivalent in both diffraction and crystal facet directions. Therefore we employ the term "crystal axis" without specifying it throughout the paper.

(8) Add explanation on the lattice symmetry (space group) of this compound, both above and below the CDW transition.

Above the CDW transition, 4Hb-TaS₂ has the P6₃/mmc hexagonal space group (#194). We added it to the description of the crystal. Below the CDW the symmetry is still under debate, as it depends on the relative stacking of the CDW distortion in the 1H and 1T layers, see for example: D.C. Miller *et al.*, "Charge density wave states in tantalum dichalcogenides", PHYSICAL REVIEW B 97, 045133 (2018).

(9) In the introduction, Sr₂RuO₄ should be added as a chiral-superconductor candidate.

Sr₂RuO₄ is a very interesting superconductor, long considered to host chiral superconductivity. However, several recent experiments cast doubt on the possibility of a chiral superconductivity, but rather point to other non-conventional order parameters. We have added it to the list of possible candidates.

(10) " H_c " in Figs. 1d, 3a, 3b, S2a, S2b, S4 should be " H_{c2} "

(11) " ΔH_c " in Figs. 3a and S2a should be " $\Delta H_c/H_{c2}$ " or something like that.

(12) For the phi-dependence data, it is better to put the ticks in the horizontal axis every 90 deg, not 100 deg, so that one can analyze the asymmetry easily.

Thank you for the comments. We have updated all the relevant figures to clearly state which critical field is discussed, and the definitions of the relative change in the critical field. In addition, we improved the horizontal ticks.

Reviewer Comments:

Reviewer #1 (Remarks to the Author):

I thank the author for providing the additional data and narrative, and for carefully considering my comments. I find the response generally satisfactory and the paper is in better shape. I still think the topic and the material is interesting, and the data is of good quality and eventually deserves to be published. I also agree with other referees that perhaps conclusions about the underlying microscopic physics picture can be better supported with additional data and/or analysis, or made less assertively if future experimental study beyond the scope of the paper is needed.

Reviewer #2 (Remarks to the Author):

The authors made some changes in their manuscript. While concerning the twofold symmetry of the CDW order to the electronic state, and the in-plane resistivity under an in-plane field with a pseudo-twofold symmetry effect, my concerns are still maintained. Thus I could not make a positive recommendation for the work.

Reviewer #3 (Remarks to the Author):

I have reviewed the manuscript NCOMMS-22-37257A "Chiral to Nematic Crossover in the Superconducting State of 4Hb-TaS₂" by I. Silber et al. resubmitted to Nature Communications.

In the first-round review, the manuscript has been reviewed by three reviewers. The first reviewer provides rather positive comments but the other two raised concerns about trivial origins of the observations. Here I only comment on the authors' response to my comments.

My main concern in the previous round was possible field misalignment (i.e. tilting of the field rotational plane with respect to the crystalline ab plane: "wobble" in the authors' wording). In my experience, for superconductors with out-of-plane anisotropy of larger than 10, experiments must be performed very very carefully to avoid any extrinsic effects. From such experience, I have raised several suggestions to experimentally deny such possibility. In particular, I requested to show data with two-axes field-direction control. Otherwise, at least, I urged data connecting the temperature region of the STM (0.38 K) and the temperature region of Hc₂ measurements. However, most of the proposals are not fully addressed, as I explain below in detail. I understand that there are several experimental difficulties and limitations. However, for high standard journals, I believe that possibility of extrinsic effects must be fully avoided. Therefore, I judge that the manuscript is not suited for publication in Nature Communications.

[A] The only added data sets are the T-dependent Hc₂-phi curves (Fig.3a) only down to 1.8 K. The new data indicate that the two-fold Hc₂ anisotropy seems to saturate when lowering temperature. Then, how this "saturation" behavior is connected to the nearly isotropic behavior at low T?

[B] I have another comment on this data. The authors argue that the T-dependent two-fold anisotropy cannot be explained by the "wobble" effect. But I don't agree. Practically, in various quasi-two-dimensional superconductors, the out-of-plane anisotropy is temperature dependent especially near zero-field T_c. For example, in Sr₂RuO₄, the anisotropy quickly drops from 60 to 20 from T = T_c to T = 0.5 T_c. Such T-dependent Hc₂ anisotropy can result in T-dependent extrinsic two-fold anisotropy due to misalignment.

[C] The only data connecting the two temperature regions is still the Hc₂ vs phi data at 0.4 K (Fig.S4). This data set remains insufficient to discuss two-fold anisotropy because it covers only the angle range

much smaller than 180-degree. The argument of very weak two-fold anisotropy at low temperature deduced from this data set is unlogical as I explained in the previous report.

[D] Related to my previous comment (5), the authors reply mentioning the average of 69 and 111 degs is not careful and logical at all. What I was asking is the reason why the two angles (69 deg and 111 deg) each have strong deviation from the expected 90 deg shifting.

[E] I have an additional comment on my previous comment (7)-(e). "in-plane crystal axis" is not unambiguous, since [110] or [210] or some other directions can be considered as a "crystal axis". I strongly recommend to specify the direction unambiguously such as "a axis" or "[100] axis".

Dr. Paul Wiecki
Associate Editor
Nature Communications

Dear Editor.

Since it is crucial for us to prove our claims and refute Referee #3 criticisms, we took it upon ourselves to perform a new set of measurements in a two-axis rotator that we recently got access to.

The new measurement procedure is described in the supplementary part, Control Experiments.

We will outline the experimental procedure here.

First, we used a flake with an estimated thickness of 200nm connected to gold pads using the e-beam lithography technique. The flake is glued on a sample mount that is inserted into a two-axis rotator probe (atto3DR piezo stack insider attoDRY2100 cryostat). The cryogenic station is equipped with a 9T magnet. The probe is equipped with an in-situ angle measurement apparatus.

Despite the precise determination of the stage angle, we first measure a possible spurious misalignment effect. Namely, we wanted to correct for changes in the out-of-plane angle when rotating in the sample plane. Since the stage angles are measured continuously, such misalignment can be due to a small angle between the sample and the stage. Such misalignment can result in spurious effects, as suggested by Referee #3. **We show that this is not the case in our experiment.**

Figure 1 (Figure S3 in the supplementary part) shows the sample's deviation from perfect perpendicular orientation during azimuthal rotation. This measurement is done by finding the out-of-plane angle at which the resistance is at a minimum close to the superconducting transition. (See Figure 1A, Figure S3a in the supplementary part). We obtain a maximum deviation of 0.2 degrees (peak to peak) for the entire azimuthal range under study. This slight deviation cannot account for the significant effects we observe. We now perform an azimuthal rotation experiment (Similar to main figure 3c), but this time correct for the deviation in the horizontal orientation. The main effect of a two-fold symmetric superconducting state is observed in both measurements (3c and S3c), but the temperature and the field ranges are different due to different cryogenic stations.

Summarizing, the two-fold nematic superconducting state is clearly observed in the two-axis rotator experiment. This effect is not due to misalignment, as suggested by Reviewer #3.

We believe that this puts an end to the Reviewer's doubts. We hope this experiment and arguments convince the Editor to publish our paper immediately without further delay.

While we believe that we refuted the main criticisms of Referee #3, we address other points raised by the Referees below.

FIG. S3. Control experiments using a two-axis rotator. (a) Raw data of resistance as a function of out-of-plane angle, at $T = 1.7$ K and $H = 7.5$ T, at various probe angles (the $\phi = 0$ direction is arbitrary here) displaying a clear minima when the field is in the plane of the sample. (b) The extracted in-plane angle as a function of the rotator angle. A minuscule change in the alignment is found for the entire range. (c) The resistance, shifted by its average, as a function of the in-plane angle at various fields at $T = 1.7$ K. Clearly, two-fold modulation of the resistance is observed. Here $\phi = 0$ was adjusted to the maximum resistance, similar to the other flake samples in the main text. The current is applied at $\phi = 80^\circ$.

Reviewer #1

We thank Reviewer #1 for their positive response.

Reviewer #2

The Reviewer did not explain what are the concerns that are still maintained. If these are the concerns raised by Reviewer #3, then with the additional new data, these concerns will hopefully be lifted.

Reviewer #3 (Remarks to the Author):

I have reviewed the manuscript NCOMMS-22-37257A "Chiral to Nematic Crossover in the Superconducting State of 4Hb-TaS2" by I. Silber et al. resubmitted to Nature Communications.

In the first-round review, the manuscript has been reviewed by three reviewers. The first Reviewer provides rather positive comments but the other two raised concerns about trivial origins of the observations. Here I only comment on the authors' response to my comments.

My main concern in the previous round was possible field misalignment (i.e. tilting of the field rotational plane with respect to the crystalline ab plane: "wobble" in the authors' wording). In my experience, for superconductors with out-of-plane anisotropy of larger than 10, experiments must be performed very very carefully to avoid any extrinsic effects. From such experience, I have raised several suggestions to experimentally deny such possibility. In particular, I requested to show data with two-axes field-direction control. Otherwise, at least, I urged data connecting the temperature region of the STM (0.38 K) and the temperature region of Hc2 measurements. However, most of the proposals are not fully addressed, as I explain below in detail. I understand that there are several experimental difficulties and limitations. However, for high standard journals, I believe that possibility of extrinsic effects must be fully avoided. Therefore, I judge that the manuscript is not suited for publication in Nature Communications.

[A] The only added data sets are the T-dependent Hc2-phi curves (Fig.3a) only down to 1.8 K. The new data indicate that the two-fold Hc2 anisotropy seems to saturate when lowering

temperature. Then, how this "saturation" behavior is connected to the nearly isotropic behavior at low T?

Unfortunately, we do not have data between 0.35K and 1.7K. This will require a new set of measurements at the magnet lab. We believe that the new data set presented here and the supplementary figure S3 shows that the anisotropy is significant, close to T_c . The low-temperature transport and the STM data suggest that the transport and the superconducting state at this temperature regime are nearly isotropic. Our experiment cannot provide any information (that is not merely speculations) about the temperature dependence between 0.35K and 1.7K.

[B] I have another comment on this data. The authors argue that the T-dependent two-fold anisotropy cannot be explained by the "wobble" effect. But I don't agree. Practically, in various quasi-two-dimensional superconductors, the out-of-plane anisotropy is temperature dependent especially near zero-field T_c . For example, in Sr_2RuO_4 , the anisotropy quickly drops from 60 to 20 from $T = T_c$ to $T = 0.5 T_c$. Such T-dependent H_{c2} anisotropy can result in T-dependent extrinsic two-fold anisotropy due to misalignment.

We have made two-axis rotator measurements clearly showing that the anisotropy is not due to wobble (See above and the supplementary part).

[C] The only data connecting the two temperature regions is still the H_{c2} vs ϕ data at 0.4 K (Fig.S4). This data set remains insufficient to discuss two-fold anisotropy because it covers only the angle range much smaller than 180 degrees. The argument of very weak two-fold anisotropy at low temperature deduced from this data set is unlogical as I explained in the previous report.

This data requires travel to the magnet lab and special equipment since the critical field is about 18 Tesla at low temperatures. We combine two probes (the STM and the low-temperature transport) that show a more isotropic superconducting state at very low temperatures. We agree that a wider temperature range would be better. However, such a scan takes a long time and requires a user facility magnet. We stress that the combination of the STM data and the transport strongly supports our conclusion that the low-temperature superconducting state is more isotropic.

[D] Related to my previous comment (5), the authors reply mentioning the average of 69 and 111 degs is not careful and logical at all. What I was asking is the reason why the two angles (69 deg and 111 deg) each have strong deviation from the expected 90 deg shifting.

This experiment is technically challenging since we had to work with two cleaves from a large single crystal where the crystal axis can be identified from a visual inspection. Such large crystals can deform in the cleaving process, smearing the superconducting features. Both parts of the cleave should survive, and the rotation between them should be under control.

[E] I have an additional comment on my previous comment (7)-(e). "in-plane crystal axis" is not unambiguous, since [110] or [210] or some other directions can be considered as a "crystal axis". I strongly recommend to specify the direction unambiguously such as "a axis" or "[100] axis".

We thank the Reviewer for the suggestion. We did as suggested.

REVIEWER COMMENTS

Reviewer #1 (Remarks to the Author):

For its overall high-quality and comprehensive experiment, I support the eventual publication of this work. However, in my previous round of review, I commented that the "conclusions about the underlying microscopic physics picture can be better supported with additional data and/or analysis, or made less assertively if future experimental study beyond the scope of the paper is needed." I don't believe my 2nd round of comments has been directly addressed, and I can't make a recommendation towards its publication until they have been. I list my remaining concerns below:

1. The author provided a two-axis rotator measurement, showing the variation of out-of-plane angle θ to be on the order of 0.2 degrees. However, the fig. S3(a) shows that $dR/d(\theta)$ is quite large near $\theta = 90$ degrees (where the dip is). Can the author quantitatively give out the R oscillation due to wobbling effect as $\Delta R = 0.2 \times dR/d(\theta)$ (while $\theta = 90$ degree), and compare to the observed oscillation magnitude of R as a function of ϕ (1.2 ohm)? I can see the trivial ΔR is smaller than 1.2 ohm, but I think it is better to elaborate quantitatively, on what portion of observed two-fold R oscillation can be attributed to realistic experimental misalignment of θ , instead of assertively excluding its contribution entirely.

2. The author acknowledges that the origin of some of the experimental observations (for example, the stripe states) is yet to be understood, yet the conclusions are made rather assertively about the microscopic mechanism of the observed 2-fold symmetry. This was one of my remaining concerns from my previous round of input, which the author did not address. Additional experiments are obviously needed to pinpoint its microscopic mechanisms, which are understandably out of the scope of this paper. As I commented in the previous round, the conclusions should be made less assertively before such future experiments provide more insights into the detailed physics picture. For example, for the final paragraph, I suggest changing:

(1) "The large variation in the critical field, of about 20%, strongly supports the existence" to ".... implies possible existence of..."

(2) "To reconcile these findings with the indications of a chiral superconducting ground state, we offer a theoretical model that captures the crossover between a chiral order parameter to a nematic one." to " these findings are consistent with a potential crossover between a chiral order parameter to a nematic one,".

(3) "Our work indicates that the pairing in 4Hb-TaS₂ evolves from a nematic, time-reversal even state ... a time-reversal breaking chiral order ..., suggesting a unique superconducting phase diagram." to "Our work demonstrates experimental signature of co-existing pairing mechanisms in 4Hb-TaS₂, which evolves as a function of temperature. Future experiment is needed to understand the detailed microscopic mechanism."

Reviewer #2 (Remarks to the Author):

The authors have revised their manuscript by adding more data with well controlled azimuthal angle between the applied field and the crystallize axis. I believe there is a slight anisotropy of either H_{c2} or the superconducting gap Δ_s . While because the anisotropy is so small, for example the H_{c2} anisotropy is around 10%, if adopting the Pippard or Ginzburg-Landau equations for an anisotropic gap in the plane, the gap anisotropy is only about 4-7%. At a temperature close to T_c , this anisotropy is more pronounced, as shown by the data in Fig.3c. Such a small gap anisotropy at low temperatures is not necessary to be expected for observation of an isotropic vortex pattern (0.38K). Thus I judge that the gap anisotropy is very small, which excludes the necessity for adopting a sophisticated model to interpret the data. The observation does not warrant the claim for the existence of two-component superconductivity or chirality of the order parameter.

Reviewer #3 (Remarks to the Author):

I have reviewed the manuscript NCOMMS-22-37257B-Z"Chiral to Nematic Crossover in the Superconducting State of 4Hb-TaS2" by I. Silber et al. resubmitted to Nature Communications.

In the second revised manuscript, the authors added important data measured under two-axis field-direction control. This new data set indicates that the two-fold behavior is not due to the field misalignment. This addresses the most important issue of my concerns. Although the connection between the low-temperature "isotropic" behavior and high-temperature nematic behavior is still somewhat uncertain, this issue may be left for future publication. Thus, I now judge that the manuscript is recommended for publication in Nature Communications.

I have a few comments on the newly added data, which might be useful for future studies.

(1) For the misalignment effect, more quantitative analysis would be possible. In Fig.S3a, the resistivity vs theta curves roughly have slopes of 1 Ohm/deg. Thus, 0.2-deg misalignment can result in at most 0.2 Ohm extrinsic signal.

(2) I suggest to pick the minimum resistance R_{min} of each curve in Fig.S3a, and plot R_{min} as a function of the in-plane angle ϕ . R_{min} must be the value when the field is exactly parallel to the plane. Thus, effects of field misalignment must be absent in such a plot.

Response letter manuscript NCOMMS-22-37257-T

Below, we address the reviewers' comments point by point:

Reviewer #1 (Remarks to the Author):

For its overall high-quality and comprehensive experiment, I support the eventual publication of this work.

We thank Reviewer #1 for their positive response and for supporting the future publication of our work.

However, in my previous round of review, I commented that the "conclusions about the underlying microscopic physics picture can be better supported with additional data and/or analysis, or made less assertively if future experimental study beyond the scope of the paper is needed." I don't believe my 2nd round of comments has been directly addressed, and I can't make a recommendation towards its publication until they have been. I list my remaining concerns below:

1. The author provided a two-axis rotator measurement, showing the variation of out-of-plane angle θ to be on the order of 0.2 degrees. However, the fig. S3(a) shows that $dR/d(\theta)$ is quite large near $\theta = 90$ degrees (where the dip is). Can the author quantitatively give out the R oscillation due to wobbling effect as $\Delta R = 0.2 \times dR/d(\theta)$ (while $\theta = 90$ degree), and compare to the observed oscillation magnitude of R as a function of ϕ (1.2 ohm)? I can see the trivial ΔR is smaller than 1.2 ohm, but I think it is better to elaborate quantitatively, on what portion of observed two-fold R oscillation can be attributed to realistic experimental misalignment of θ , instead of assertively excluding its contribution entirely.

We are sorry that our description of the experimental procedure was not clear enough. After measuring the variation of out-of-plane angle θ (figure S3B), we found a 0.2 degree variation. This means that the plane of the sample is not perfectly aligned with the plane of the probe. We then accounted for this 0.2 degrees variation (figure 3SC) and zeroed the out-of-plane variations. This results in no more than a 0.01 degree wobble. Therefore, there is no point in calculating the ΔR as a function of the out-of-plane variation.

We have clarified this point further in the "Control Experiments" supplementary part.

2. The author acknowledges that the origin of some of the experimental observations (for example, the stripe states) is yet to be understood, yet the conclusions are made rather assertively about the microscopic mechanism of the observed 2-fold symmetry. This was one of my remaining concerns from my previous round of input, which the author did not address. Additional experiments are obviously needed to pinpoint its microscopic mechanisms, which are understandably out of the scope of this paper. As I commented in the previous round, the conclusions should be made less assertively before such future experiments provide more insights into the detailed physics picture. For example, for the final paragraph, I suggest changing:

(1) "The large variation in the critical field, of about 20%, strongly supports the existence" to "... implies possible existence of..."

We thank the Reviewer for the suggestion, which we have implemented. We hope our paper will inspire additional experiments on this fascinating material.

(2) "To reconcile these findings with the indications of a chiral superconducting ground state, we offer a theoretical model that captures the crossover between a chiral order parameter to a nematic one. " to " these findings are consistent with a potential crossover between a chiral order parameter to a nematic one,".

We have implemented the Reviewer's suggestion.

(3) *"Our work indicates that the pairing in 4Hb-TaS2 evolves from a nematic, time-reversal even state ... a time-reversal breaking chiral order ..., suggesting a unique superconducting phase diagram."* to *"Our work demonstrates experimental signature of co-existing pairing mechanisms in 4Hb-TaS2, which evolves as a function of temperature. Future experiment is needed to understand the detailed microscopic mechanism."*

We thank the Reviewer, and we have implemented the Reviewer's suggestion and removed this sentence.

Reviewer #2 (Remarks to the Author):

The authors have revised their manuscript by adding more data with well controlled azimuthal angle between the applied field and the crystallize axis. I believe there is a slight anisotropy of either H_{c2} or the superconducting gap Δ_s . While because the anisotropy is so small, for example the H_{c2} anisotropy is around 10%, if adopting the Pippard or Ginzburg-Landau equations for an anisotropic gap in the plane, the gap anisotropy is only about 4-7%. At a temperature close to T_c , this anisotropy is more pronounced, as shown by the data in Fig.3c. Such a small gap anisotropy at low temperatures is not necessary to be expected for observation of an isotropic vortex pattern (0.38K). Thus I judge that the gap anisotropy is very small, which excludes the necessity for adopting a sophisticated model to interpret the data. The observation does not warrant the claim for the existence of two-component superconductivity or chirality of the order parameter.

We agree with the Reviewer that the superconductivity in 4Hb-TaS2 is anisotropic in-plane. Our STM resolution permits discernment of approximately 2-4% coherence length anisotropy at 0.4K, precluding the detection of smaller anisotropies. A possible model could be that the superconducting anisotropy diminishes with temperature until it lowers below our detection limit.

Importantly, our model, while not minimal to our results, aims to encompass both the observed in-plane anisotropic superconductivity and our previous findings indicating spontaneous time-reversal symmetry breaking appearing together with superconductivity. This suggests a chiral ground state (References 21). We have added a disclaimer to acknowledge the departure from minimal considerations for consistency with prior research.

Reviewer #3 (Remarks to the Author):

I have reviewed the manuscript NCOMMS-22-37257B-Z, "Chiral to Nematic Crossover in the Superconducting State of 4Hb-TaS2" by I. Silber et al. resubmitted to Nature Communications.

In the second revised manuscript, the authors added important data measured under two-axis field-direction control. This new data set indicates that the two-fold behavior is not due to the field misalignment.

This addresses the most important issue of my concerns.

Although the connection between the low-temperature "isotropic" behavior and high-temperature nematic behavior is still somewhat uncertain, this issue may be left for future publication. Thus, I now judge that the manuscript is recommended for publication in Nature Communications.

We thank the Reviewer for considering the new data and recommending it for publication.

I have a few comments on the newly added data, which might be useful for future studies.

(1) For the misalignment effect, more quantitative analysis would be possible. In Fig.S3a, the resistivity vs theta curves roughly have slopes of 1 Ohm/deg. Thus, 0.2-deg misalignment can result in at most 0.2 Ohm extrinsic signal.

We kindly refer the Reviewer to our response to Reviewer #1, in which we discuss the fact that we used the two-axis rotator to, in fact, zero the misalignment of 0.2 degree.

(2) I suggest to pick the minimum resistance R_{min} of each curve in Fig.S3a, and plot R_{min} as a function of the in-plane angle ϕ . R_{min} must be the value when the field is exactly parallel to the plane. Thus, effects of field misalignment must be absent in such a plot.

We thank the Reviewer for their suggestion. First, we plot the minimal resistance from Fig.S3a as a function of the in-plane angle, when the in-plane is defined zero for maximal resistance:

Next, we compare this analysis to our carefully aligned in-plane measurement (Fig. S3c, the same field of 7.5T), as explained in our response to Reviewer #1:

Clearly, the two methods are consistent, demonstrating that we have indeed zeroed the wobble.